# Building a community of practice through social media using the hashtag #neoEBM

Amy Keir[1,2,3]*, Nicolas Bamat[4,5], Bron Hennebry[2,3], Brian King[6], Ravi Patel[7], Clyde Wright[8], Alexandra Scrivens[9], Omar ElKhateeb[10], Souvik Mitra[11], Damian Roland[12,13]

1 SAHMRI Women and Kids, South Australian Health and Medical Research Institute, Adelaide, South Australia, Australia, 2 Adelaide Medical School and the Robinson Research Institute, The University of Adelaide, Adelaide, South Australia, Australia, 3 MedSTAR Emergency Retrieval Service, SA Ambulance Service, South Australia, Australia, 4 Division of Neonatology and Center for Pediatric Clinical Effectiveness, Children's Hospital of Philadelphia, Philadelphia, PA, United States of America, 5 Department of Pediatrics, Perelman School of Medicine at the University of Pennsylvania, Philadelphia, PA, United States of America, 6 Department of Pediatrics, Section of Neonatology, Baylor College of Medicine, Houston, TX, United States of America, 7 Emory University School of Medicine and Children's Healthcare of Atlanta, Atlanta, Georgia, United States of America, 8 Department of Pediatrics, Section of Neonatology, Children's Hospital Colorado and The School of Medicine, The University of Colorado, Denver, CO, United States of America, 9 Newborn Services, John Radcliffe Hospital, Oxford University Hospitals, NHS Foundation Trust, Oxford, United Kingdom, 10 King Fahad Medical City, Riyadh, Saudi Arabia, 11 Departments of Pediatrics, Community Health & Epidemiology, Dalhousie University, Halifax, Nova Scotia, Canada, 12 Children's Emergency Department, Paediatric Emergency Medicine Leicester Academic (PEMLA) Group, Leicester Royal Infirmary, Leicester, United Kingdom, 13 SAPPHIRE Group, Health Sciences, Leicester University, Leicester, United Kingdom

* amy.keir@adelaide.edu.au

**Data Availability Statement:** The minimal dataset underlying this study is available within the manuscript and its Supporting Information files, and from the University of Adelaide Figshare (https://doi.org/10.25909/14329754.v1). Additional

## Abstract

### Objectives

Social media use is associated with developing communities of practice that promote the rapid exchange of information across traditional institutional and geographical boundaries faster than previously possible. We aimed to describe and share our experience using #neoEBM (Neonatal Evidence Based Medicine) hashtag to organise and build a digital community of neonatal care practice.

### Materials and methods

Analysis of #neoEBM Twitter data in the Symplur Signals database between 1 May 2018 to 9 January 2021. Data on tweets containing the #neoEBM hashtag were analysed using online analytical tools, including the total number of tweets and user engagement.

### Results

Since its registration, a total of 3 228 distinct individual Twitter users used the hashtag with 23 939 tweets and 37 259 710 impressions generated. The two days with the greatest number of tweets containing #neoEBM were 8 May 2018 (n = 218) and 28 April 2019 (n = 340), coinciding with the annual Pediatric Academic Societies meeting. The majority of Twitter users made one tweet using #neoEBM (n = 1078), followed by two tweets (n = 411) and

data are available from the Symplur Signals database and can be accessed through www.symplur.com with a fee-based account subscribed for the hashtag #neoEBM. The authors did not have any special access to the data that other researchers would not have.

**Funding:** AK receives funding from the Australian National Health and Medical Research Council (NHMRC) www.nhmrc.gov.au (APP1161379). The contents of this paper are solely the responsibility of the individual authors and do not reflect the views of the NHMRC. The funder had no role in study design, data collection and analysis, decision to publish, or preparation of the manuscript.

**Competing interests:** AK, RP, CW, SM, AS, OE and BK are current or previous social media editors and NB is immediate past lead social media editor for the International Society for Evidence-Based Neonatology (ebneo.org) and all tweet as part of these roles as @EBNEO. None are compensated for their role as social media editors. The authors have no financial relationships with the online analytical platform Symplur. This does not alter our adherence to PLOS ONE policies on sharing data and materials.

more than 10 tweets (n = 347). The number of individual impressions (views) of tweets containing #neoEBM was 37 259 710. Of the 23 939 tweets using #neoEBM, 17 817 (74%) were retweeted (shared), 15 643 (65%) included at least one link and 1 196 (5%) had at least one reply. As #neoEBM users increased over time, so did tweets containing #neoEBM, with each additional user of the hashtag associated with a mean increase in 7.8 (95% CI 7.7–8.0) tweets containing #neoEBM.

## Conclusion

Our findings support the observation that the #neoEBM community possesses many of the characteristics of a community of practice, and it may be an effective tool to disseminate research findings. By sharing our experiences, we hope to encourage others to engage with or build online digital communities of practice to share knowledge and build collaborative networks across disciplines, institutions and countries.

## Introduction

A community of practice (CoP) is a group of people who share a concern or a passion for something they do, and learn how to do it better through regular interaction [1]. Social media use is associated with the development of CoPs as it promotes the transfer of information at scale and crosses institutional and geographical boundaries faster than previously possible. These forums allow shared conversations around awareness and critical appraisal of new research evidence that is transparent, accessible to users and timely [1].

The timely implementation of medical research evidence into clinical practice is an important public health challenge, with the evidence-to-practice gap famously described as taking up to 17 years [2]. New approaches to narrow this gap are urgently needed, and online CoPs may be a valuable tool in an era of growing digital interaction [3, 4]. An attempt to organise and build a CoP for those involved in neonatal care was made in May 2018 by an international group of physicians following the Pediatric Academic Societies (PAS) meeting in Toronto, Canada. The group meeting idea originated on Twitter and coordinated via the same platform. Collectively, all members of this group were already using the social media platform Twitter to disseminate new research findings relevant to neonatal practice.

Twitter is an interactive social media microblogging platform established in 2006 that allows users to send 280-character messages or links, known as tweets, to each other. Hashtags are words or phrases preceded by the hash or pound sign (#) and used on social media platforms to classify digital content as relating to a specific topic. As a result of the meeting at PAS in 2018, the hashtag #neoEBM was selected to potentially disseminate and track evidence-based neonatology content [5]. This observational cohort study aims to describe the use of #neoEBM over time and the impact of this social-media based intervention on the evolution of an online CoP among neonatal healthcare professionals and stakeholders on Twitter.

## Materials and methods

Our study was a descriptive, observational cohort study utilising the social media health analytic website Symplur as the data source. User generated posts ("tweets") met inclusion criteria if they were generated on Twitter and contained the hashtag #neoEBM between 1 May 2018 and 9 January 2021. All other tweets were excluded. This was a convenience sample of all

eligible posts without formal sample size calculation. The hashtag was registered on 8 May 2018 on Symplur (www.symplur.com/submit-hashtag) allowing assessment of the use and impact of posts containing #neoEBM over time. We utilised the NOECO [6] statement, a standardised framework for reporting social media analytics, to hypothesise that on the digital platform Twitter (*network*) the use of the hashtag #neoEBM (*object*) led to the establishment of a community of practice (*observation*) using the network analysis system Symplur (*engine*). Symplur is an online tool integrated into the Twitter microblogging service and assimilates data on specific health-related hashtags. We purchased access to the database for one month through the Symplur Signals Self-Serve (research option for 10 datasets) option. We created a Symplur Signals database to analyse tweets and impressions (user views) that contained the registered #neoEBM hashtag. Data on the number of individual Twitter users using #neoEBM, the total number of tweets using #neoEBM (excluding retweets), impressions (user views) of #neoEBM and retweets (re-posting another users' Tweet) were obtained. User description (e.g., neonatologist, medical journal, etc.) and global location were obtained when publicly available in the profile.

The study was approved by the University of Adelaide's Low-Risk Human Research Ethics Review (HREC) Group (Faculty of Health and Medical Sciences) Approval No. H-2020-003. The HREC waived the requirement for individual consent. The data collection for this study was compliant with the terms and conditions of both Twitter and Symplur.

## Results

The Symplur Signals database was interrogated from 1 May 2018 to 9 January 2021 (2 years and 8 months of data; 981 days) on 10 January 2021. Since the registration of the #neoEBM hashtag in May 2018, a total of 3 228 distinct Twitter users utilised the hashtag #neoEBM at least once (Figs 1 and 2).

The location of Twitter users included the United States of America (n = 494; 15%), United Kingdom (n = 467; 15%), Spain (n = 110, 3%) and Australia (n = 103; 3%). The 20 most frequent users of #neoEBM hashtag were neonatal providers or other stakeholders in neonatal

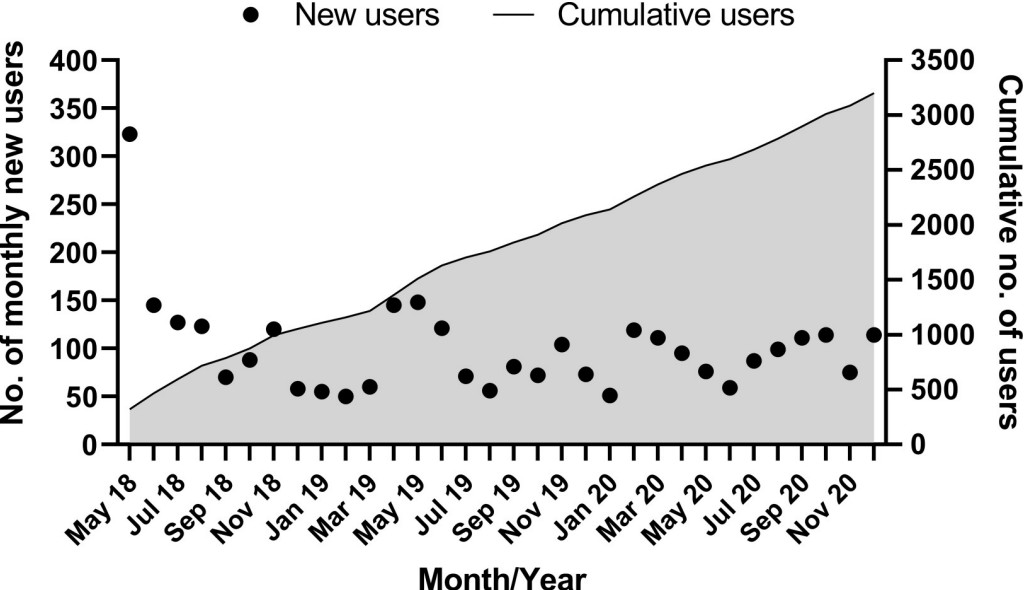

**Fig 1. Cumulative users of the hashtag #neoEBM over time.**

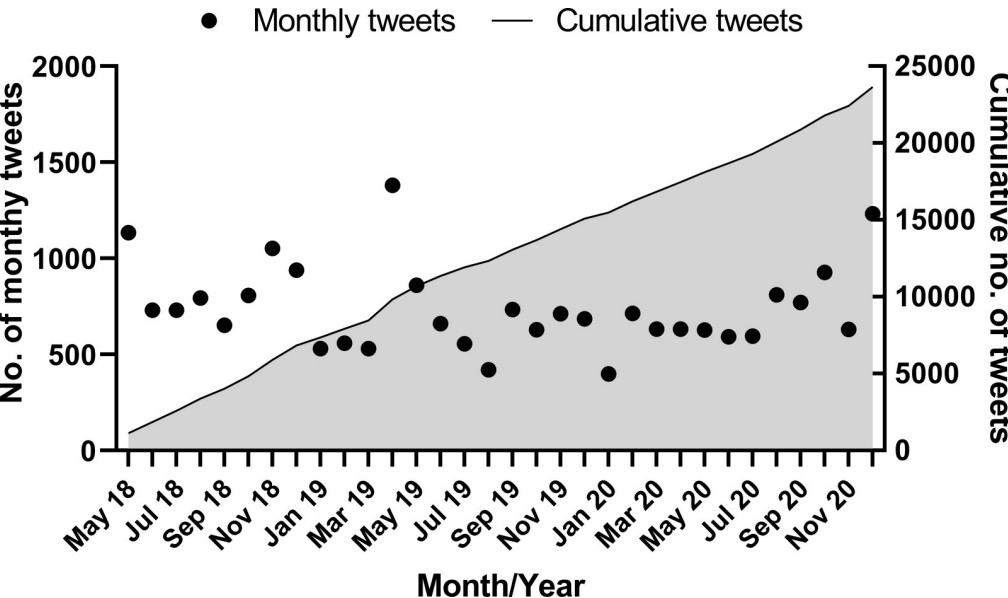

**Fig 2. Cumulative tweets of the hashtag #neoEBM over time.**

research. These were neonatologists (n = 11), a neonatal fellow (n = 1), neonatal nurses (n = 2), peer-reviewed medical journals (n = 2), neonatal research/advocacy groups (n = 3) and one bot (a bot is a software application used to automatically generate messages and act as a follower of users on social media platforms) developed by a paediatric resident (n = 1) to retweet posts containing #neoEBM. Cumulatively, these top 20 users have a following of 35 361 accounts with an average of 1768 each with a range 152–8 827.

There have been 23 939 unique tweets using #neoEBM. The two days with the most significant number of tweets containing #neoEBM were Tuesday 8 May 2018 (n = 218) and Sunday 28 April 2019 (n = 340), both coinciding with the annual Pediatric Academic Societies meeting (Fig 3).

The majority of Twitter users made one tweet using #neoEBM (n = 1843), followed by two tweets (n = 443) and more than 10 tweets (n = 347). The number of individual impressions (views) of tweets containing #neoEBM was 37 259 710.

Of the 23 939 tweets using #neoEBM, 17 817 (74%) were retweeted (shared), 15 643 (65%) included at least one link and 1 196 (5%) had at least one reply. There is an average of 100 new users of #neoEBM per month (Fig 3).

As #neoEBM users increased over time, so did tweets containing #neoEBM, with each additional user of the hashtag associated with a mean increase in 7.8 (95% CI 7.7–8.0) tweets containing #neoEBM (Fig 4).

The top 10 words used in tweets containing #neoEBM were infants (n = 4 209), preterm (n = 2 885), neonatal (n = 2 354), study (n = 2 384), published (n = 1 551), outcomes (n = 1 326), trial (n = 1 438), use (n = 1 215), review (n = 1 181) and risk (n = 1 180).

A comparison of the core components of a CoP and the characteristics of the #neoEBM community are displayed in Table 1.

## Discussion

Our study provides preliminary evidence that the #neoEBM community contains the critical components of a CoP refined and expanded by Aveling in Table 1 [7]. These components

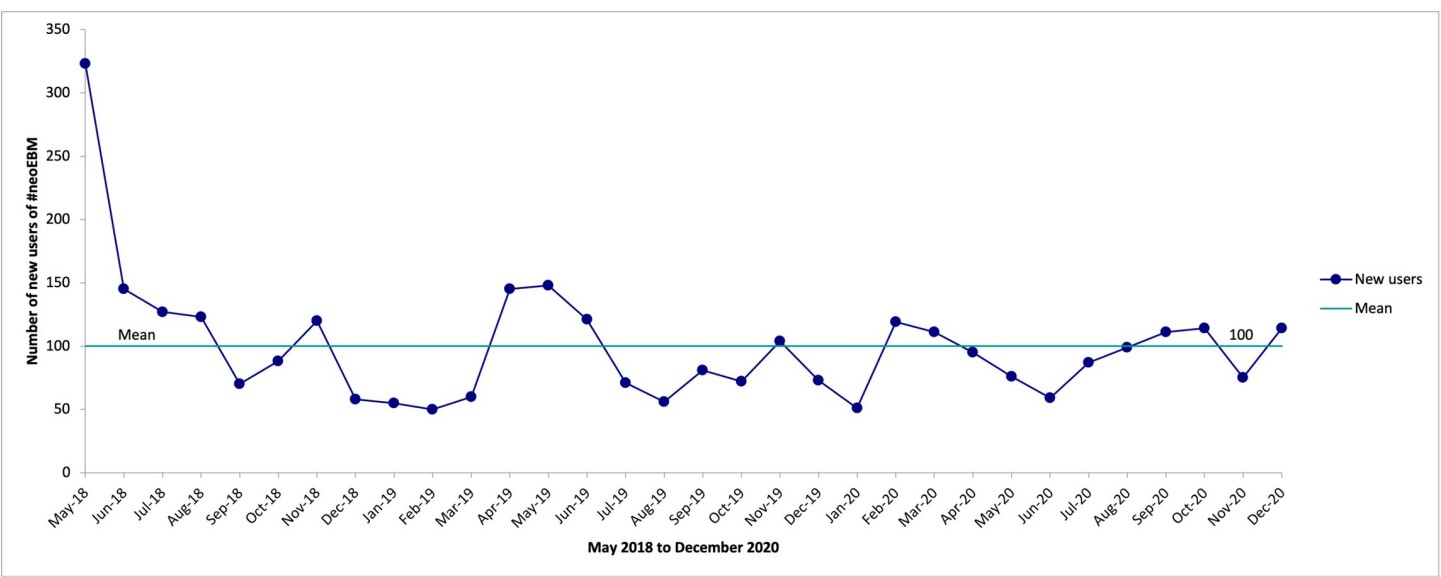

**Fig 3. Time-series graph of numbers of new #neoEBM users (monthly average).**

include a community formed by interdependent groups and individuals from different professions and organisations, united by a common purpose of bridging the gap between neonatal research evidence and current clinical practice, coming together to build, share and disseminate knowledge. Additional components include exploiting a social media platform for knowledge diffusion, operation through vertical and horizontal structures, deployment of peer influence, usage of informal mechanisms to achieve change and harnessing community power to seek solutions to the challenges of knowledge dissemination [7]. This suggests those using the #neoEBM hashtag may be part of the development of an online CoP in neonatal care as a way to share knowledge with each other. This online CoP offers an open digital space for information sharing, with a flat hierarchy, strong group identity, high engagement (involvement), and rapid flow of information and knowledge translation consistent with other online

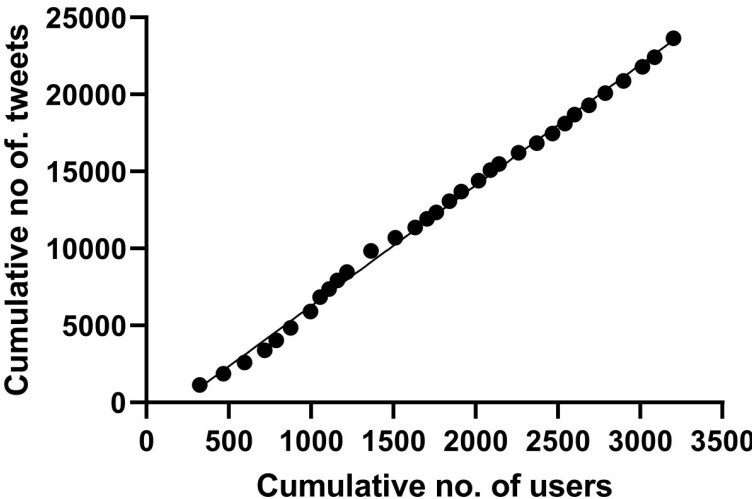

**Fig 4. Cumulative numbers of tweets and users over time.**

**Table 1. Aveling's core components of clinical communities of practice and alignment with the #neoEBM community.**

| Aveling's core components of clinical communities of practice | #neoEBM community |
|---|---|
| Interdependent groups and individuals | #neoEBM hashtag connects individuals who demonstrate interactions with each other |
| Members cross clinical and organisational boundaries | |
| | Top 20 Twitter users of #neoEBM include: Neonatologists (n = 11) |
| | Neonatal fellow (n = 1) |
| | Neonatal nurses (n = 2) |
| | Peer-reviewed medical journals (n = 2) |
| | Neonatal research/advocacy groups (n = 3) |
| | Bot tweeting #neoEBM (n = 1) |
| | Users from various academic and medical organisations across the world participate. |
| | Further details on these 20 users are available in the S1 Data. |
| Members are united by a common purpose of bridging the gap between best research evidence and current clinical practice | Content of #neoEBM is focused on healthcare-related themes, centred on research study findings and access to neonatal research content |
| Members come together not only to learn or share knowledge but to achieve those aims | Physical gatherings were responsible for the development and adoption of #neoEBM, publicized by the original Tweet t.co/xtcjQYotM7 and demonstrated by the ongoing use of #neoEBM. Physical gatherings by users have continued at subsequent academic meetings. |
| Exploits the networks' inherent potential for effective and low-cost knowledge generation and diffusion | #neoEBM is centred on an effort to diffuse new research evidence to promote knowledge generation at scale among neonatal stakeholders, and generates subnetworks around individuals, e.g., specific websites |
| Operates through both vertical and lateral structures | The network expands through increasing individuals who influence others across increasingly broad geographic areas |
| | Demonstrated by #neoEBM users located in the United States of America (n = 494; 15%), United Kingdom (n = 467; 15%), Spain (n = 110, 3%) and Australia (n = 103; 3%). |
| Harnesses the power of the community and its collective wisdom when seeking solutions to problems | Interactions (measured through mentions) expand rather than contract over time |
| | Demonstrated by increasing numbers of users of #neoEBM (Fig 1) |
| Deploys peer influence and uses primarily informal, social control mechanisms to achieve change | Most frequent users (i.e., top 20 users described above) exert influence, but this changes over time |

healthcare CoPs [8]. Our findings are supported by the development of other online CoPs in healthcare [3, 4] and provide the foundation for using this CoP to impact healthcare quality in neonatal care [7]. The increases seen (Fig 3) in the use of #neoEBM around major paediatric conferences (Pediatric Academic Societies Annual Meeting) reflects the role of the #neoEBM hashtag as a tool to promote knowledge exchange. Academic conferences routinely serve as an opportunity to share new, impactful research evidence. The temporal association between these meetings and increased activity among the CoP is unlikely to be coincidence, but rather reflect the occurrence of new knowledge in need of sharing. Though it may seem paradoxical to observe increased digital activity within a CoP during a time of physical gathering, this highlights the important role that digital interaction plays in modern CoP, continuing to serve as a medium for communication despite physical proximity.

Limitations to our study include the inherent challenges to confirm alterations in clinical practices by CoP members. However, there is a suggestion through social media posts that the use of #neoEBM has allowed for faster dissemination of study findings that have changed practice (e.g., https://bit.ly/2ZiLjOq).

Potential criticism of our work includes that it is no different to an email listserv, that it is at high risk of being overtaken by opinion-based medicine and that it has or will have no impact on clinical practice. The size of the #neoEBM community and its sustainment suggest engagement through an evidence hierarchy that reaches beyond perceived or reported benefits. With the evidence-to-practice gap famously described as up to 17 years [2], different approaches to our current ones to close this gap are urgently needed. Synoptic, curated and accessible educational material for healthcare professionals [9] is an excellent place to start. Social media is widely accessible in resource-poor settings, whereas journals and email listservs are intrinsically exclusive and therefore often inaccessible in these settings.

Roger's diffusion of innovations theory is often used to explain the spread of new ideas and practices in a wide variety of settings, including in healthcare [10]. It provides a framework for analysing the process of innovation as it unfolds to make more informed choices and decisions about how to guide the dissemination and diffusion of these new ideas [11]. The #neoEBM community is striving to enter into the 'early majority' phase to legitimise our innovation to bring us up to the described tipping point of 'acceptance' at 15–20% adoption. We are aware that the late majority are sceptical, perhaps reflected by the expressed criticism received of our work, and laggards put trust in the status quo. We will continue to question the status quo and reflect on Donald M Berwick's statement on leading the improvement of systems that "*Effective leaders challenge the status quo both by insisting that the current system cannot remain and by offering clear ideas about superior alternatives*" [12]. We want the #neoEBM community to be a place for anyone working in or interested in neonatal care, including families and former patients, to learn and share knowledge about new research findings.

While we have demonstrated elements of organisation, the impact of the CoP on the practice of individual clinicians is yet to be determined. Further qualitative and quantitative study is needed to understand how practice may change through participation in the #neoEBM community. It cannot also be assumed that all CoPs will add value in the same way and further research is needed in different specialties and professions to see how widespread this approach is in healthcare.

## Conclusions

The #neoEBM community possesses many of the characteristics of a CoP and appears to promote knowledge exchange within the community. By sharing our experiences, we hope to encourage others to engage with or build online digital CoPs to share knowledge and build collaborative networks across disciplines, institutions and countries. Further challenges for our emerging digital CoP include sustaining it into the future, the need to evaluate further the potential impact of #neoEBM on clinical practice and convince the 'late majority' that social media is a useful and accessible tool for evidence dissemination.

## Supporting information

**S1 Data.**
(XLSX)

## Author Contributions

**Conceptualization:** Amy Keir, Nicolas Bamat, Ravi Patel, Clyde Wright, Omar ElKhateeb, Damian Roland.

**Data curation:** Amy Keir, Nicolas Bamat, Brian King, Ravi Patel, Clyde Wright, Alexandra Scrivens, Omar ElKhateeb, Damian Roland.

**Formal analysis:** Amy Keir, Ravi Patel, Damian Roland.

**Funding acquisition:** Amy Keir.

**Investigation:** Nicolas Bamat, Clyde Wright, Omar ElKhateeb, Souvik Mitra, Damian Roland.

**Methodology:** Amy Keir, Nicolas Bamat, Ravi Patel, Damian Roland.

**Resources:** Damian Roland.

**Validation:** Amy Keir, Nicolas Bamat, Bron Hennebry, Brian King, Ravi Patel, Clyde Wright, Alexandra Scrivens, Souvik Mitra, Damian Roland.

**Writing – original draft:** Amy Keir.

**Writing – review & editing:** Amy Keir, Nicolas Bamat, Bron Hennebry, Brian King, Ravi Patel, Clyde Wright, Alexandra Scrivens, Omar ElKhateeb, Souvik Mitra, Damian Roland.

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
