## [Decision Letter · Decision Letter 0]

19 Feb 2021

PONE-D-21-01208

#neoEBM: Building a community of practice through social media

PLOS ONE

Dear Dr. Keir,

Thank you for submitting your manuscript to PLOS ONE. After careful consideration, we feel that it has merit but does not fully meet PLOS ONE’s publication criteria as it currently stands. Therefore, we invite you to submit a revised version of the manuscript that addresses the points raised during the review process.

The reviewers have highlighted several issues concerning the current status of the paper. The paper needs MAJOR revisions in order to be evaluated for a future publication. I suggest you to follow more in detail each suggestion.

We look forward to receiving your revised manuscript.

Kind regards,

Barbara Guidi

Academic Editor

PLOS ONE

Journal Requirements:

2. In your Methods section, please include additional information about your dataset and ensure that you have included a statement specifying whether the collection method complied with the terms and conditions for the website.

3. In your ethics statement in the Methods section and in the online submission form, please confirm that all data were fully anonymized before you accessed them and/or whether the IRB or ethics committee waived the requirement for informed consent.

"AK, RP, CW, SM, AS, OE and BK are current or previous social media editors and NB is immediate past lead social media editor for the International Society for Evidence-Based Neonatology (ebneo.org) and all tweet as part of these roles as @EBNEO. None are compensated for their role as social media editors. The authors have no financial relationships with the online analytical platform Symplur."

Reviewers' comments:

Reviewer's Responses to Questions

**Comments to the Author**

1. Is the manuscript technically sound, and do the data support the conclusions?

Reviewer #1: Yes

Reviewer #2: Partly

Reviewer #3: No

Reviewer #4: Yes

2. Has the statistical analysis been performed appropriately and rigorously? 

Reviewer #1: Yes

Reviewer #2: No

Reviewer #3: No

Reviewer #4: Yes

3. Have the authors made all data underlying the findings in their manuscript fully available?

Reviewer #1: Yes

Reviewer #2: Yes

Reviewer #3: No

Reviewer #4: No

4. Is the manuscript presented in an intelligible fashion and written in standard English?

Reviewer #1: Yes

Reviewer #2: No

Reviewer #3: Yes

Reviewer #4: Yes

5. Review Comments to the Author

Reviewer #1: Reviewer comments :

Building a community of practice through social media

The study was interesting to go through. These are my main concerns. Below comments follow the order of the manuscript.

Title

I recommend adding to the title as it is not clear enough (too broad).

Introduction

The authors should add to the introduction a paragraph on the importance of the practice and gap in knowledge. Also, they need to add some literature in this regard. Other comments in introduction part is indicated as below:

Line 65: the authors should change the numbering of the reference to number 5 into 2, so it will follow sequential order.

Line 70: please delete the word meeting, as it’s a repetition.

Line 73: start a new Paragraph-Twitter is an….

Line 78: Please clarify the type of the observational study conducted.

Material and Methods

-Lines 83-85: the study design should be clarified.

-Sample size and sample size calculation.

-Where there any exclusion criteria?

Results

Figure 1: the writing in the figure is not clear.

Table 1: Aveling’score- please edit the space between s c.

Discussion

-This should be supported by other studies in different medical fields that show an effect of social media on Cop.

-Need recommendations for further studies and what should be done.

Thank you

Reviewer #2: This is an article discussing on the use of social media to develop a community of practice. I appreciate the initiatives on such social media-based research and saw merits of this paper and should be considered for publication upon major corrections.

My comments and suggestions: The authors shall improve the manuscripts (also to serve the need of readers of PLOS ONE of different backgrounds, unlike medical journal of specialised readers). In brief, the manuscript needs to be better organised and provide further elaboration to support the ideas put forward in the manuscript as in the following areas:

Introduction:

The numbering of the reference is not according to the sequence. Perhaps, also some background on why or the driver of online CoP, perhaps from other established fields.

Materials and methods:

Some narration on NOECO by Roland et al shall be included to make your methods clearer.

Results:

I feel that all data reported by authors should be analyzed and interpreted further, rather than to report as is. For example, the authors reported the cumulative users over time; however, data is only slightly described by the authors in the section of engagement metrics. Interpretation such as the trend of users and tweets, projection in future numbers, the number of users with different number of tweets could add values to the manuscript.

I expect the information such as the locations and twitters can also be further described and interpreted, rather than simply reported the data. (For instance, rhe authors reported from selected countries with numbers, and what about the rest, since the readers will be curious to know, for instance, from a total of 3228, US, UK, Spain and Australia were 494, 467, 110 and 103, what about the other locations of the remaining 2054?)

Of course, the figures attached must be in good resolution, and the detailed information is difficult to read.

Table 1 compared the core components of a CoP and the characteristics of community. This is the key finding that #neoEBM community fits into the 8 features by Aveling. The information in this table shall be further explained by the authors. For example, the top 20 twitter users were identified by the authors, and they were said to be originated from interdependent groups and individuals, and the members cross clinical and organizational boundaries. How does this determination made by the authors were not described and it should be supported by the background data.

As for the content (second row), the authors stated that the twitter users were members united for a common purpose and come to achieve the same aim, how do the authors come to such determination shall be discussed and supported by the analysis. The same also goes to other points in Table 1. A further description on the data will benefit the readers for better understanding.

Discussion:

There were several issues which need attention by the authors. How the authors determined the identity of twitter users, the issue of robotic tweets, potential of mis-hashtag by the users, confidentiality of the tweeted message, issue of the authorship on tweeted information, and etc are factors that potentially affect the interpretation of the data.

Also, I expect the authors to provide some view how the findings will lead to a greater quality improvement (QI) in clinical community in general.

Reference:

The list needs some minor formatting.

Reviewer #3: In my opinion, the study titled “#neoEBM: Building a community practice through social media” is interesting but has major limitations:

Introduction: A clear rationale for the current study along with use of tools such as “Aveling’s core component of clinical community” is missing.

Materials and methods: missing inclusion exclusion criteria for evaluating the dependent variables. This is vaguely presented but the actual process must be clearly presented.

The results section needs significant work. The figure axes are not legible. The quality of figure content is very poor. Additionally, the authors need to elaborate on the results and their relation to the presented figures.

Figure 2: it is expected that the frequency of tweets would be higher around scientific events as highlighted by the authors. But it is important that the authors elaborate on what this means and involves in their discussion.

Discussion needs significant work on the above factors.

In the conclusion, the authors make a case that #neoEBM is an effective tool to disseminate research findings via social media platform (twitter in this case). What percentage of the research community would this reach? Are there any analyses to project the percentage of population it would reach compared to the standard Pubmed, google scholar and other scholar manager approaches? Is the information through #noEBM from twitter reliable? The significance and impact of these findings need to be discussed.

Finally, the data were not easily accessible to review.

Reviewer #4: Thank you for giving me the opportunity to review this paper. This paper addresses an interesting topic of building a community through Twitter, in order to increase the dissemination of research recommendation. The idea is great, the team had done a good job in building this community. However, several information was missing. The comments below are intended to further strengthen the paper.

Title

1- The title could be improved by removing the hashtag, to be "Building a community of practice through social media".

Abstract

1- Line 57, can you please provide the long form of any abbreviation when you mention in it the first time, please?

Introduction

1- It was not clear about the context in which the hashtag was introduced. Was there any previous attempt to engage people using for example, a mailing list? Did all the group twitter users get any formal training for using twitter when it was introduced? were there anyone in charge of the hashtag? How many hours per week was spent in observing the hashtag? Was there any previous attempt to assess the utilization of the hashtag?

2- Line 65, can you please add the references numbers based on their appearance in the paper? As the introduction line 63 started with reference number 1, and line 65, the second appeared reference number is 5. Can you please clarify if this should be number 2?

3- Can you please provide a references to the statements in lines 74-76.

4- Can you please provide a definition for the word hashtag, line 77.

Materials and methods

1- Line 83, Can you please comment on the stage of hashtag registration please? Was this completed to assess the analysis using the Symplur software, or was this needed by Twitter, to register official hashtag?

2- Line 85, can you please confirm what is NOECO, please?

3- Line 88, can you please confirm if the "this online tool" refers to Symplur?

4- Line 108, 36% of user's demographics is presented in the paper, can you please confirm the location of the other users please?

5- Line 110, please clarify what is meant by top users?

6- Line 112, can you please define the word bot, please?

7- Can you please comment on the total number of tweets if all of them were about neonatology? Was all the tweets were read/ a sample of the tweets were assessed as part of quality assurance to confirm that the tweets are about neonatology? Or if they were include based on face value, that all tweets were about neonatology? Or were there any user used the hashtag to tweet about another topic?

8- Can you please confirm if there were any duplicated tweets?

9- Can you please provide the definition of the word engagement, please?

10- Have the team considered completing a content or thematic analysis of the tweets?

11- Can you please confirm if you had completed a word association thematic analysis, please?

12- Can you please comment on the analysis that was completed using the software, Symplur, please?

13- Can you please add the information mentioned the price of the software in the method section, please?

Results

1- Line 120-121, can you please clarify the numbers please, as the numbers are not the same as the numbers presented in the abstract line 47-48. Can you please clarify if the numbers should add up to 3,228, please?

2- Can you please clarify the content of the tweets? What are the main themes within the tweets?

3- Can you please confirm if the tweets were used for information dissemination, increase awareness about certain topics, or networking? And what is the percentage of each component?

4- What are the demographics of people tweeting using the hashtag? What was the gender of the people?

5- Can you please confirm what was the average engagement rate, please?

6- Can you please confirm if important information about the characterises of the tweets were factored in the analysis including, the length of the tweets, the time tweet was posted, the number of links in the tweets, and if these variables affected the engagement rate?

Discussion

1- Line 160-161, can you please comment that this statement is based on an observation from 3 reply to one tweet, please?

6. PLOS authors have the option to publish the peer review history of their article (what does this mean?). If published, this will include your full peer review and any attached files.

Reviewer #1: No

Reviewer #2: No

Reviewer #3: No

Reviewer #4: No

---

## [Author Response · Author response to Decision Letter 0]

30 Mar 2021

Thank you again for the detailed and very helpful comments from all the reviewers. Please find our detailed responses below as well as in the marked up copy of the revised paper:

Reviewer #1: Reviewer comments:

Building a community of practice through social media. The study was interesting to go through. These are my main concerns. Below comments follow the order of the manuscript.

Comment 1: 

Title

I recommend adding to the title as it is not clear enough (too broad).

Reply: We have altered the title to “Building a community of practice through social media using the hashtag #neoEBM”

Comment 2:

Introduction

The authors should add to the introduction a paragraph on the importance of the practice and gap in knowledge. Also, they need to add some literature in this regard. 

Reply: Additional information about the evidence-to-practice gap is included now and referenced.

Comment 3:

Other comments in introduction part is indicated as below:

Line 65: the authors should change the numbering of the reference to number 5 into 2, so it will follow sequential order.

Reply: There was a referencing error and reference 5 has been corrected.

Line 70: please delete the word meeting, as it’s a repetition. 

Reply: Deleted

Line 73: start a new Paragraph-Twitter is an….

Reply: New paragraph started.

Comment 4:

Line 78: Please clarify the type of the observational study conducted.

Reply: Cohort now included.

Comment 5:

Material and Methods

-Lines 83-85: the study design should be clarified.

-Sample size and sample size calculation.

-Where there any exclusion criteria?

Reply: We have included the following information in the Material and Methods section:

Our study was a descriptive, observational cohort study utilising the social media health analytic website Symplur as the data source. User generated posts (“tweets”) met inclusion criteria if they were generated on Twitter and contained the hashtag #neoEBM between 1 May 2018 and 9 January 2021. All other tweets were excluded. This was a convenience sample of all eligible posts without formal sample size calculation.

Comment 6:

Results

Figure 1: the writing in the figure is not clear.

Table 1: Aveling’score- please edit the space between s c.

Reply: Our apologies, the figure has been updated and the typographical error corrected.

Comment 7:

Discussion

-This should be supported by other studies in different medical fields that show an effect of social media on Cop.

Reply: Thank you for this comment, we have added further information and relevant citations into the discussion:

“Our findings are supported by the development of other online CoPs in healthcare.7, 8”

Comment 8:

-Need recommendations for further studies and what should be done.

Reply: “While we have demonstrated elements of organisation, the impact of the CoP on the practice of individual clinicians is yet to be determined. Further qualitative and quantitative study is needed to understand how practice may change through participation in the #neoEBM community. It cannot also be assumed that all CoP will add value in the same way and further research is needed in different specialties and professions to see how widespread this approach is in healthcare.”

Reviewer #2: This is an article discussing on the use of social media to develop a community of practice. I appreciate the initiatives on such social media-based research and saw merits of this paper and should be considered for publication upon major corrections.

My comments and suggestions: The authors shall improve the manuscripts (also to serve the need of readers of PLOS ONE of different backgrounds, unlike medical journal of specialised readers). In brief, the manuscript needs to be better organised and provide further elaboration to support the ideas put forward in the manuscript as in the following areas:

Comment 1:

Introduction:

The numbering of the reference is not according to the sequence. 

Reply: Thank you for this comment, we have corrected our referencing.

Comment 2: Perhaps, also some background on why or the driver of online CoP, perhaps from other established fields.

Reply: We have now included the following in the introduction/background:

The timely implementation of medical research evidence into clinical practice is an important public health challenge, with the evidence-to-practice gap famously described as taking up to 17 years.2 New approaches to narrow this gap are urgently needed, and online CoPs may be a valuable tool in an era of growing digital interaction.

Comment 3:

Materials and methods:

Some narration on NOECO by Roland et al shall be included to make your methods clearer.

Reply: We have now included a brief explanation around NOECO in the methods section.

Comment 4:

Results:

I feel that all data reported by authors should be analyzed and interpreted further, rather than to report as is. For example, the authors reported the cumulative users over time; however, data is only slightly described by the authors in the section of engagement metrics. Interpretation such as the trend of users and tweets, projection in future numbers, the number of users with different number of tweets could add values to the manuscript.

Reply: We appreciate the request for further detail in the analysis, but we are currently limited by information provided by third party platforms. We have provided information commensurate with previous research in this area (reference 8) and hope that this work will enable future research groups, including ours, to push to expand data sources to enhance the information available. 

Comment 5:

I expect the information such as the locations and twitters can also be further described and interpreted, rather than simply reported the data. (For instance, rhe authors reported from selected countries with numbers, and what about the rest, since the readers will be curious to know, for instance, from a total of 3228, US, UK, Spain and Australia were 494, 467, 110 and 103, what about the other locations of the remaining 2054?)

Reply: We have added language to the Methods to clarify that the location of the user is only available when publicly shared by the user and updated to Results to describe the proportion of users lacking this information. Unfortunately, no further information of the location much beyond what is supplied in the paper is available – the majority of users’ locations are deemed ‘unknown’. 

Comment 6:

Of course, the figures attached must be in good resolution, and the detailed information is difficult to read.

Reply: Please accept our apologies, we have updated the figures to a higher resolution and reviewed them closely for legibility. These are now formatted in TIFF format at 600 dpi per PLOS One guidance. 

Comment 7:

Table 1 compared the core components of a CoP and the characteristics of community. This is the key finding that #neoEBM community fits into the 8 features by Aveling. The information in this table shall be further explained by the authors. For example, the top 20 twitter users were identified by the authors, and they were said to be originated from interdependent groups and individuals, and the members cross clinical and organizational boundaries. How does this determination made by the authors were not described and it should be supported by the background data?

Reply: Further detailed data on the individual Twitter users identified within this group is now provided in the Supplementary Information.

Comment 8:

As for the content (second row), the authors stated that the twitter users were members united for a common purpose and come to achieve the same aim, how do the authors come to such determination shall be discussed and supported by the analysis. The same also goes to other points in Table 1. A further description on the data will benefit the readers for better understanding.

Reply: We have included additional information in the Supplementary Material and ensured that there is a clear justification for each assertion in Table 1.

Comment 9:

Discussion:

There were several issues which need attention by the authors. How the authors determined the identity of twitter users, the issue of robotic tweets, potential of mis-hashtag by the users, confidentiality of the tweeted message, issue of the authorship on tweeted information, and etc are factors that potentially affect the interpretation of the data.

Reply: 

Twitter users were identified by the use of the #neoEBM hashtag. It would be possible for some of the users to have been ‘bots’ however these numbers are likely to be small because (i) NeoEBM is a very niche hashtag and there is little or no commercial value and (ii) the authors of the study have lived experience in twitter and are able to recognise individuals outside of social media norms. Twitter is a public forum in which you sign an agreement at the outset that your tweets will be made public unless you specifically specify them to be private. Access to this data set means all tweets have been made publicly available. Finally, while it is possible that people were tweeting on behalf of other users this would have to have occurred at a massive scale to have impacted on the results, especially given the homogeneity shown in the content themes, and so we feel this is of limited concern. 

Comment 10:

Also, I expect the authors to provide some view how the findings will lead to a greater quality improvement (QI) in clinical community in general.

Reply: 

We hope that our work is a facilitator for the change processes needed at collective, organisational and cultural levels to improve the use of evidence in everyday clinical practice. We have briefly touched on and cited the following paper in the response to the reviewer’s comment:

Aveling EL, Martin G, Armstrong N, Banerjee J, Dixon-Woods M. Quality improvement through clinical communities: eight lessons for practice. J Health Organ Manag. 2012;26(2):158-74. doi: 10.1108/14777261211230754. PMID: 22856174.

Comment 11:

Reference:

The list needs some minor formatting.

Reply: Thank you, we have done this.

 

Reviewer #3: 

In my opinion, the study titled “#neoEBM: Building a community practice through social media” is interesting but has major limitations:

Comment 1:

Introduction: A clear rationale for the current study along with use of tools such as “Aveling’s core component of clinical community” is missing.

Reply: Thank you for this comment. We have more clearly written in the introduction that there is evidence (referenced) that highlights social media communities on twitter may act as communities of practice. There is therefore a theoretical link between the development of emerging communities of practice and improvements in care (although the extent of this link needs further evaluation). Communities of practice have previously not been described in neonatal medicine and this study sought to examine this. 

Comment 2:

Materials and methods: missing inclusion exclusion criteria for evaluating the dependent variables. This is vaguely presented but the actual process must be clearly presented.

Reply: We have included (in addition to more detail about the study design in response to reviewers’ other comments), the following:

“Our study was a descriptive, observational cohort study utilising the social media health analytic website Symplur as the data source. User generated posts (“tweets”) met inclusion criteria if they were generated on Twitter and contained the hashtag #neoEBM between 1 May 2018 and 9 January 2021. All other tweets were excluded. This was a convenience sample of all eligible posts without formal sample size calculation.”

Comment 3:

The results section needs significant work. The figure axes are not legible. The quality of figure content is very poor. Additionally, the authors need to elaborate on the results and their relation to the presented figures.

Reply: The figures have been updated to be legible (again, please accept our apologies for this) and we have adjusted the axis labelling to improve legibility. These are formatted in TIFF format at 600 dpi per PLOS One guidance. Please also see response below.

Comment 4:

Figure 2: it is expected that the frequency of tweets would be higher around scientific events as highlighted by the authors. But it is important that the authors elaborate on what this means and involves in their discussion.

Reply: Thank you for this comment and this demonstrates that our findings mimic real-life observations (thus validating our work) and we have included in the discussion the following “highlight how communities of practice benefit from face-to-face interactions at conferences.” 

Comment 5:

Discussion needs significant work on the above factors.

Reply: Please see our responses and additions to the paper in response to the above (and other significant additions to the discussion).

Comment 6:

In the conclusion, the authors make a case that #neoEBM is an effective tool to disseminate research findings via social media platform (twitter in this case). What percentage of the research community would this reach? Are there any analyses to project the percentage of population it would reach compared to the standard Pubmed, google scholar and other scholar manager approaches? Is the information through #noEBM from twitter reliable? The significance and impact of these findings need to be discussed.

Reply: Further details regarding the additional limitations to our work has been added to the discussion. We have also included further information in the discussion around future research directions that hopefully also addresses this comment.

Comment 7:

Finally, the data were not easily accessible to review.

Reply: We have included our basic dataset as Supplementary Information.

 

Reviewer #4: 

Thank you for giving me the opportunity to review this paper. This paper addresses an interesting topic of building a community through Twitter, in order to increase the dissemination of research recommendation. The idea is great, the team had done a good job in building this community. However, several information was missing. The comments below are intended to further strengthen the paper.

Comment 1:

Title

1- The title could be improved by removing the hashtag, to be "Building a community of practice through social media".

Reply: Thank you for this comment. It does conflict with another reviewer’s comment to make the title more specific. 

At this point we have altered it to state:

“Building a community of practice through social media using the hashtag #neoEBM”. However, we would be happy to alter it again as needed.

Comment 2:

Abstract

1- Line 57, can you please provide the long form of any abbreviation when you mention in it the first time, please?

Reply: We have removed the abbreviation and used the long-form.

Comment 3:

Introduction

1- It was not clear about the context in which the hashtag was introduced. Was there any previous attempt to engage people using for example, a mailing list? Did all the group twitter users get any formal training for using twitter when it was introduced? were there anyone in charge of the hashtag? How many hours per week was spent in observing the hashtag? Was there any previous attempt to assess the utilization of the hashtag?

Reply: The formation of #neoEBM was a spontaneous event by clinicians already engaged and knowledgeable about twitter. The reason this started was that the group were keen to more dissemination of research beyond email lists, and we were all familiar with twitter. 

By definition, there is no one in charge of hashtag, as this is the way in which an organic community of practice develops. Some previous work on this has been published (reference 5: Keir et al, EBM_BMJ 2019).

Comment 4:

2- Line 65, can you please add the references numbers based on their appearance in the paper? As the introduction line 63 started with reference number 1, and line 65, the second appeared reference number is 5. Can you please clarify if this should be number 2?

Reply: Our apologies, there was an error in the referencing, and we have corrected it.

Comment 5:

3- Can you please provide a references to the statements in lines 74-76.

Reply: We have referenced these statements.

Comment 6:

4- Can you please provide a definition for the word hashtag, line 77.

Reply: We have provided a definition for the word hashtag accordingly.

Comment 7:

Materials and methods

1- Line 83, Can you please comment on the stage of hashtag registration please? Was this completed to assess the analysis using the Symplur software, or was this needed by Twitter, to register official hashtag?

Reply: We registered the hashtag #neoEBM within 3 days of our meeting at PAS in 2018 with the concept to monitor its usage over time using the Symplur software. Registration was required by Symplur to allow us to use their analytics platform.

Comment 8:

2- Line 85, can you please confirm what is NOECO, please?

Reply: Please see our reply to Reviewer 2, Comment 3. It is a standardised framework for reporting social media analytics research.

Comment 9:

3- Line 88, can you please confirm if the "this online tool" refers to Symplur?

Reply: Yes, it does, we have made it clearer in the paper.

Comment 10:

4- Line 108, 36% of user's demographics is presented in the paper, can you please confirm the location of the other users please?

Reply: Unfortunately, it is not possible to define the rest of the users’ locations as the majority of users’ locations are ‘unknown’. We have explained this more clearly in the update methods section.

Comment 11:

5- Line 110, please clarify what is meant by top users?

Reply: We have clarified this by including the definition of ‘top’:

“The top 20 Twitter users of #neoEBM, defined as the most frequent users of the hashtag,….”

Comment 12:

6- Line 112, can you please define the word bot, please?

Reply: A bot is a software application used to automatically generate messages, advocate ideas, act as a follower of users on social media platforms. We have elaborated on this definition when the term is introduced in the results section. 

Comment 13:

7- Can you please comment on the total number of tweets if all of them were about neonatology? Was all the tweets were read/ a sample of the tweets were assessed as part of quality assurance to confirm that the tweets are about neonatology? Or if they were include based on face value, that all tweets were about neonatology? Or were there any user used the` hashtag to tweet about another topic?

Reply: Not all tweets were individually read but a word cloud highlights the content were universally related to neonatal practice (we can provide the word cloud if necessary). We can find no evidence that #NeoEBM is used by any other group or community 

Comment 14:

8- Can you please confirm if there were any duplicated tweets?

Reply: There were no duplicated tweets – all were unique.

Comment 15:

9- Can you please provide the definition of the word engagement, please?

Reply: We are using the word engagement to mean “being involved with something (aka the CoP).” We have highlighted this in the revised paper by including the word “involvement” next to engagement to clarify its intended use in the paper.

Comment 16:

10- Have the team considered completing a content or thematic analysis of the tweets?

Reply: Yes, thank you for this suggestion. As this is significant piece of work, we think it would be better placed as a further piece of work.

Comment 17:

11- Can you please confirm if you had completed a word association thematic analysis, please?

Reply: Thank you for this query. No, we have not done this and would like to do this as an additional piece of work. Please see our response to the above.

Comment 18:

12- Can you please comment on the analysis that was completed using the software, Symplur, please?

Reply: Symplur is a twitter analytics platform which has developed trademarked algorithms to analyse content and also uses a taxonomy of 35000 terms mapped to 1 million social profiles which are then broken down into relevant healthcare stakeholders. Their software collates information around a specific hashtag and allows for deep content analysis via national language processing algorithms. 

Comment 19:

13- Can you please add the information mentioned the price of the software in the method section, please?

Reply: We have added the following:

”We paid US$499 for access to the database for one month through the Symplur Signals Self-Serve (research option for 10 datasets) option.”

Comment 20:

Results

1- Line 120-121, can you please clarify the numbers please, as the numbers are not the same as the numbers presented in the abstract line 47-48. Can you please clarify if the numbers should add up to 3,228, please?

Reply: The number of users of #neoEBM across the study period was 3 228, this is consistent is the abstract and result sections.

To further clarify: “The majority of Twitter users made one tweet using #neoEBM (n=1843), followed by two tweets (n=443) and more than 10 tweets (n=347).” These numbers do not add up to 3 228 as some users made between 3-9 tweets and are not included in this breakdown of numbers.

We have corrected our error in the abstract and the main text. Thank you for picking this up.

Comment 21:

2- Can you please clarify the content of the tweets? What are the main themes within the tweets?

Reply: Please see our reply to Comment 13 (Reviewer 4) and previous comments about a thematic analysis.

Comment 22:

3- Can you please confirm if the tweets were used for information dissemination, increase awareness about certain topics, or networking? And what is the percentage of each component?

Reply: Thank you for this query. Unfortunately, it is not possible to draw inferences from the data regarding these points unfortunately due to the limitations of the third-party software currently available.

Comment 23:

4- What are the demographics of people tweeting using the hashtag? What was the gender of the people?

Reply: We have provided some of this data in the details regarding the top 20 users and in the Supplementary Material. Further, we have clarified in the methods that characteristics of users were obtained when publicly available in the profile. 

Comment 24:

5- Can you please confirm what was the average engagement rate, please?

Reply: This was not formally calculated as part of this project. Our rationale was that as it is not part of the Aveling’s principles, and we did not formally calculate this. 

Comment 25:

6- Can you please confirm if important information about the characterises of the tweets were factored in the analysis including, the length of the tweets, the time tweet was posted, the number of links in the tweets, and if these variables affected the engagement rate?

Reply: Thank you for these comments. hese factors were not factored into the analysis. Our rationale was that as it is not part of the Aveling’s principles, and we did not formally calculate this.

Comment 26:

Discussion

1- Line 160-161, can you please comment that this statement is based on an observation from 3 reply to one tweet, please?

Reply: Thank you for this comment. These statements were meant to highlight the potential criticisms of our work. Consequently, we have re-worded the sentence to hopefully make this more clear:

“Potential criticism of our work includes that it is no different to an email listserv, that it is at high risk of being overtaken by opinion-based medicine and that it has or will have no impact on clinical practice.”

6. PLOS authors have the option to publish the peer review history of their article (what does this mean?). If published, this will include your full peer review and any attached files.

Do you want your identity to be public for this peer review? For information about this choice, including consent withdrawal, please see our Privacy Policy.

Reviewer #1: No

Reviewer #2: No

Reviewer #3: No

Reviewer #4: No

---

## [Decision Letter · Decision Letter 1]

28 Apr 2021

PONE-D-21-01208R1

Building a community of practice through social media using the hashtag #neoEBM

PLOS ONE

Dear Dr. Keir,

Thank you for submitting your manuscript to PLOS ONE. After careful consideration, we feel that it has merit but does not fully meet PLOS ONE’s publication criteria as it currently stands. Therefore, we invite you to submit a revised version of the manuscript that addresses the points raised during the review process.

The paper needs a MINOR REVISION. Please follow the suggestion given by the reviewers in order to improve the readability of the paper.

We look forward to receiving your revised manuscript.

Kind regards,

Barbara Guidi

Academic Editor

PLOS ONE

Journal Requirements:

Reviewers' comments:

Reviewer's Responses to Questions

**Comments to the Author**

1. If the authors have adequately addressed your comments raised in a previous round of review and you feel that this manuscript is now acceptable for publication, you may indicate that here to bypass the “Comments to the Author” section, enter your conflict of interest statement in the “Confidential to Editor” section, and submit your "Accept" recommendation.

Reviewer #1: All comments have been addressed

Reviewer #2: All comments have been addressed

2. Is the manuscript technically sound, and do the data support the conclusions?

Reviewer #1: Yes

Reviewer #2: Yes

3. Has the statistical analysis been performed appropriately and rigorously? 

Reviewer #1: Yes

Reviewer #2: Yes

4. Have the authors made all data underlying the findings in their manuscript fully available?

Reviewer #1: Yes

Reviewer #2: Yes

5. Is the manuscript presented in an intelligible fashion and written in standard English?

Reviewer #1: Yes

Reviewer #2: Yes

6. Review Comments to the Author

Reviewer #1: Thank you for the detailed comments. No additional comments are available for now. The author addressed majority of the reviewers comments which improved from the paper status.

Reviewer #2: The authors have made significant improvement to the manuscript. The concerns that I raised earlier were appropriately taken into consideration, or with justification to why certain recommendations were not able to be included into the text.

Some minor things that the authors should take note:

Line 67 - should the reference in a square bracket?

Line 77 - a quick search on internet shows that twitter has 330 million accounts - ref 4 might not be an updated source.

Line 167 and Line 207 - the authors stated that #neoEBM as "a tool to disseminate new knowledge" and "disseminate research findings", respectively... this makes the platform to be deemed as a "one way traffic" communication, i.e. from those who know to those who don't, or to those who have conducted research to those who haven't..one-way flow of new information. I would prefer to use "a tool to share knowledge, or to promote knowledge exchange... (i.e. the element of two-way communication) simply because knowledge presented by someone might not necessarily be "new" to everyone. Also, not all member in #neoEBM are researchers, I think. I feel that it is important to point this out, though it is more like my personal opinion. The original concept of CoP by Wenger (1991) is to serve as a platform for novices to meet experts, and if you define CoP as such, it is ok to see that this platform disseminates "new" knowledge and "research findings. However, the authors define CoP based on [1], i.e. a broader and a newer definition of CoP as "a platform to support members to interact with each other, share knowledge, promote knowledge exchange, and build the sense of belonging to the group". .. (see abstract), and therefore it will be more inclusive for the authors to use proper wordings such as share or exchange.

Line 201: Should "all C" be "all CoPs"?

7. PLOS authors have the option to publish the peer review history of their article (what does this mean?). If published, this will include your full peer review and any attached files.

Reviewer #1: No

Reviewer #2: No

---

## [Author Response · Author response to Decision Letter 1]

9 May 2021

Thank you to the reviewers for their feedback and please find our detailed responses below:

Reviewer #1: Thank you for the detailed comments. No additional comments are available for now. The author addressed majority of the reviewers’ comments which improved from the paper status.

Reply: Thank you.

Reviewer #2: The authors have made significant improvement to the manuscript. The concerns that I raised earlier were appropriately taken into consideration, or with justification to why certain recommendations were not able to be included into the text. Some minor things that the authors should take note:

Comment 1:

Line 67 - should the reference in a square bracket?

Reply: Yes, thank you and updated.

Comment 2: 

Line 77 - a quick search on internet shows that twitter has 330 million accounts - ref 4 might not be an updated source.

Reply: Thank you for pointing this out. We have decided to remove this sentence as it will clearly get more and more out of date as time goes on and is general background only.

Comment 3:

Line 167 and Line 207 - the authors stated that #neoEBM as "a tool to disseminate new knowledge" and "disseminate research findings", respectively... this makes the platform to be deemed as a "one way traffic" communication, i.e. from those who know to those who don't, or to those who have conducted research to those who haven't..one-way flow of new information. I would prefer to use "a tool to share knowledge, or to promote knowledge exchange... (i.e. the element of two-way communication) simply because knowledge presented by someone might not necessarily be "new" to everyone. 

Reply: We have updated both areas with the new (improved) wording – thank you.

Comment 4:

Also, not all member in #neoEBM are researchers, I think. I feel that it is important to point this out, though it is more like my personal opinion. The original concept of CoP by Wenger (1991) is to serve as a platform for novices to meet experts, and if you define CoP as such, it is ok to see that this platform disseminates "new" knowledge and "research findings. However, the authors define CoP based on [1], i.e. a broader and a newer definition of CoP as "a platform to support members to interact with each other, share knowledge, promote knowledge exchange, and build the sense of belonging to the group". .. (see abstract), and therefore it will be more inclusive for the authors to use proper wordings such as share or exchange. 

Reply: Thank you for this point – we agree – we hope that our CoP is inclusive and have added this point in the paper in several places. Please see the marked up version of the paper for further details, including at line 200, where we have added:

We want the #neoEBM community to be a place for anyone working in or interested in neonatal care, including families and former patients, to learn and share knowledge about new research findings.

Comment 5: Line 201: Should "all C" be "all CoPs"?

Reply: Yes, thanks for picking this up and we have corrected it.

---

## [Decision Letter · Decision Letter 2]

17 May 2021

Building a community of practice through social media using the hashtag #neoEBM

PONE-D-21-01208R2

Dear Dr. Keir,

We’re pleased to inform you that your manuscript has been judged scientifically suitable for publication and will be formally accepted for publication once it meets all outstanding technical requirements.

Kind regards,

Barbara Guidi

Academic Editor

PLOS ONE

Additional Editor Comments (optional):

Reviewers' comments:

Reviewer's Responses to Questions

**Comments to the Author**

1. If the authors have adequately addressed your comments raised in a previous round of review and you feel that this manuscript is now acceptable for publication, you may indicate that here to bypass the “Comments to the Author” section, enter your conflict of interest statement in the “Confidential to Editor” section, and submit your "Accept" recommendation.

Reviewer #1: All comments have been addressed

Reviewer #2: All comments have been addressed

2. Is the manuscript technically sound, and do the data support the conclusions?

Reviewer #1: Yes

Reviewer #2: Yes

3. Has the statistical analysis been performed appropriately and rigorously? 

Reviewer #1: Yes

Reviewer #2: Yes

4. Have the authors made all data underlying the findings in their manuscript fully available?

Reviewer #1: Yes

Reviewer #2: Yes

5. Is the manuscript presented in an intelligible fashion and written in standard English?

Reviewer #1: Yes

Reviewer #2: Yes

6. Review Comments to the Author

Reviewer #1: The authors have addressed majority of issues raised and the manuscript have improved. No additional comments are

available for now.

Reviewer #2: Thank you for the response. I feel that it is now suitable for consideration to be published. It is hope that the authors would continue monitor the development of the platform and, perhaps, update the readers in a future publication.

7. PLOS authors have the option to publish the peer review history of their article (what does this mean?). If published, this will include your full peer review and any attached files.

Reviewer #1: **Yes: **Sara AL-Musharaf

Reviewer #2: No

---

## [Editor Report · Acceptance letter]

19 May 2021

PONE-D-21-01208R2 

Building a community of practice through social media using the hashtag #neoEBM 

Dear Dr. Keir:

I'm pleased to inform you that your manuscript has been deemed suitable for publication in PLOS ONE. Congratulations! Your manuscript is now with our production department. 

Kind regards, 

on behalf of

Dr. Barbara Guidi 

Academic Editor

PLOS ONE